# SEQUENCE TO SEQUENCE TRANSDUCTION WITH HARD MONOTONIC ATTENTION

**Roee Aharoni & Yoav Goldberg**
Computer Science Department
Bar-Ilan University
Ramat-Gan, Israel
`{roee.aharoni,yoav.goldberg}@gmail.com`

## ABSTRACT

We present a supervised sequence to sequence transduction model with a hard attention mechanism which combines the more traditional statistical alignment methods with the power of recurrent neural networks. We evaluate the model on the task of morphological inflection generation and show that it provides state of the art results in various setups compared to the previous neural and non-neural approaches. Eventually we present an analysis of the learned representations for both hard and soft attention models, shedding light on the features such models extract in order to solve the task.

## 1 INTRODUCTION

Neural sequence to sequence transduction became a prominent approach to natural language processing tasks like morphological inflection generation (Faruqui et al., 2016) and automatic summarization (Rush et al., 2015) among others. A common way to improve the vanilla encoder-decoder framework for sequence to sequence tasks is the (soft) attention mechanism (Bahdanau et al., 2014) which enables the decoder to attend at specific elements in the encoded sequence, overcoming the issues in encoding very long sequences to a single vector.

It was also shown that the attention mechanism effectively learns an alignment between the input and the output sequences which is naturally found in the data, and practically learns this alignment to attend at the relevant elements of the input. However, in many NLP tasks like automatic transliteration or morphological inflection generation, the data is roughly monotonically aligned – meaning the ability of the soft attention model to attend at the entire input sequence may be sub-optimal for such tasks, while also requiring a relatively large amount of training examples which is not always available (especially for morphologically rich, low resource languages).

There have been several works on neural sequence transduction with monotonic assumptions. One approach is to train an alignment-aware RNN-based transducer which is composed of two independent RNN's – one over the input sequence and one over the output sequence. The output distribution is computed by feeding a pair of aligned RNN states through an MLP, where the alignment is defined by null symbols in the output (Graves, 2012) or by a parameterized transition probability (Yu et al., 2016). In both cases training is performed by marginalizing over all possible alignments using a forward-backward procedure. This approach lacks an attention mechanism, as a dependency between the input and output RNN's would make the inference intractable. Other related approaches employ modifications to the soft-attention mechanism, like attending on a fixed sized window over the input sequence (Jaitly et al., 2015) or "smoothing" and "sharpening" the soft-attention weight distribution in different manners (Chorowski et al., 2015). These works are motivated by the need to attend over the very long input sequences found in speech recognition. We suggest that for shorter input sequences like the characters of a word in natural language, a simple, hard attention mechanism over the elements of a bi-directional encoder may be sufficient.

More traditional approaches to monotonic sequence transduction in the NLP literature were hand-engineered finite state transducers (FST) (Koskenniemi, 1983; Kaplan & Kay, 1994) which relied on expert knowledge, or weighted finite state transducers (Mohri et al., 1997; Eisner, 2002) which combined expert knowledge with data-driven parameter tuning. While the FST approaches may

work well even on small datasets due to their engineered structure, it may be cumbersome to use them while conditioning on the entire output history as it requires a very large set of states, resulting in a model conditioning only on the last predicted output symbol (Rastogi et al., 2016).

We propose a model which handles the above issues by directly modeling a monotonic alignment between the input and output sequences which is used to perform hard attention. The model consists of an encoder-decoder neural network with a dedicated control mechanism: in each step, the decoder is fed with a single attended input state and either writes a symbol to the output sequence or advances the attention pointer to the next input state from the bi-directionally encoded sequence, as described visually in Figure 1.

This modeling suits the natural monotonic alignment between the input and output very well, as the network learns to attend at the relevant inputs before writing the output which they are aligned to. A bi-directional encoder together with the hard attention mechanism enables to condition on the entire input sequence, as each element in the input is represented using a concatenation of a forward LSTM and a backward LSTM over the input sequence. Since each element representation is aware of the entire context, non-monotone relations are also captured, which is important in tasks where segments in the output sequence are a result of long range dependencies in the input sequence. The recurrent nature of the decoder, together with a dedicated feedback connection that passes the last prediction to the next decoder step explicitly, enables the model to also condition on the entire output history at each prediction step. The hard attention mechanism allows the network to jointly align and transduce while using a focused representation at each step, rather then the weighted sum of representations used in the soft attention model. A simple training procedure using independently learned alignments enables training the network with correct alignments from the first gradient-based update, using a convenient cross-entropy loss.

To evaluate our model, we perform extensive experiments on three previously studied datasets for the morphological inflection generation task, which involves generating a target word (e.g. "härtestem", the German word for "hardest"), given a source word (e.g. "hart", the German word for "hard") and the morpho-syntactic attributes of the target (POS=adjective, gender=masculine, type=superlative etc.). Several studies showed that inflection generation is beneficial for phrase-based machine translation (Chahuneau et al., 2013) and more recently for neural machine translation (García-Martínez et al., 2016). We show that while our model is on par or better than the previous neural and non-neural state-of-the-art models on the task, it is also performing significantly better for very small training sets, being the first neural model to surpass the performance of a weighted FST model with latent variables specifically tailored for the task (Dreyer et al., 2008). Finally, we analyze and compare our model and the soft attention model, showing how they function very similarly with respect to the alignments and representations they learn, in spite of our model being much simpler.

## 2 THE HARD ATTENTION MODEL

### 2.1 MOTIVATION

We would like to transduce the input sequence, $x_{1:n} \in \Sigma_x^*$ into the output sequence, $y_{1:m} \in \Sigma_y^*$, where $\Sigma_x$ and $\Sigma_y$ are the input and output vocabularies, respectively. Imagine a machine with read-only, random access to the encoding of the input sequence, and a single pointer that determines the current read location. We can then model the sequence transduction as a series of write operations and pointer movement operations. In the case where the alignment between the sequences is monotonic, the pointer movement can be controlled by a single "move one step forward" operation (step) which we add to the output vocabulary. We implement this behavior using an encoder-decoder neural network, with a control mechanism which determines in each step of the decoder whether it is time to predict an output symbol or promote the attention pointer the next element of the encoded input.

### 2.2 MODEL DEFINITION

In prediction time, we seek the output sequence $y_{1:m} \in \Sigma_y^*$, for which:

$$y_{1:m} = \arg\max_{y'} p(y'|x_{1:n}, f) \tag{1}$$

Where: $x \in \Sigma_x^*$ is the input sequence and: $f = \{f_1, ..., f_m\}$ is a set of features influencing the transduction task (for example, in the inflection generation task these would be the desired morpho-syntactic features of the output sequence). Since we want our model to force a monotonic alignment between the input and the output, we instead look for a sequence of actions: $s_{1:q} \in \Sigma_s^*$, where: $\Sigma_s = \Sigma_y \cup \{step\}$. This sequence is the step/write action sequence required to go from $x_{1:n}$ to $y_{1:m}$ according to the monotonic alignment between them. In this case we define:

$$s_{1:q} = \arg\max_{s'} p(s'|x_{1:n}, f) = \arg\max_{s'} \prod_{s_i' \in s'} p(s_i'|s_0'...s_{i-1}', x_{1:n}, f) \qquad (2)$$

We can then estimate this using a neural network:

$$s_{1:q} = \arg\max_{s'} NN(x_{1:n}, f, \Theta) \qquad (3)$$

Where the network's parameters $\Theta$ are learned using a set of training examples. We will now describe the network architecture.

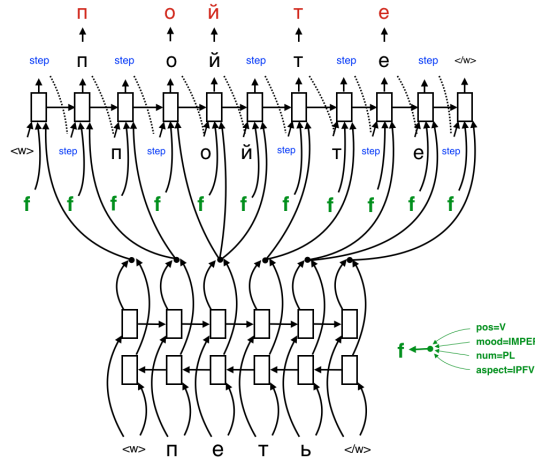

Figure 1: The hard attention network architecture. A round tip expresses concatenation of the inputs it receives. The attention is promoted to the next input element once a step action is predicted.

## 2.3 NETWORK ARCHITECTURE

**Notation** We use bold letters for vectors and matrices. We treat LSTM as a parameterized function $\mathbf{LSTM}_\theta(\mathbf{x}_1...\mathbf{x}_n)$ mapping a sequence of input vectors $\mathbf{x}_1...\mathbf{x}_n$ to a an output vector $\mathbf{h}_n$.

**Encoder** For every element in the input sequence: $x_{1:n} = x_1...x_n$, we take the corresponding embedding: $\mathbf{e}_{x_1}...\mathbf{e}_{x_n}$, where: $\mathbf{e}_{x_i} \in \mathbb{R}^E$. These embeddings are parameters of the model which will be learned during training. We then feed the embeddings into a bi-directional LSTM encoder (Graves & Schmidhuber, 2005) which results in a sequence of vectors: $\mathbf{x}_{1:n} = \mathbf{x}_1...\mathbf{x}_n$, where each vector: $\mathbf{x}_i = [\mathbf{LSTM_{forward}}(\mathbf{e}_{x_1}, \mathbf{e}_{x_2}, ...\mathbf{e}_{x_i}), \mathbf{LSTM_{backward}}(\mathbf{e}_{x_n}, \mathbf{e}_{x_{n-1}}...\mathbf{e}_{x_i})] \in \mathbb{R}^{2H}$ is a concatenation of the forward LSTM and the backward LSTM outputs when fed with $\mathbf{e}_{x_i}$.

**Decoder** Once the input sequence is encoded, we feed the decoder RNN, $\mathbf{LSTM_{dec}}$, with three inputs at each step:

1. The current attended input, $\mathbf{x}_a \in \mathbb{R}^{2H}$, initialized with the first element of the encoded sequence, $\mathbf{x}_1$.

2. A set of feature embeddings that influence the generation process, concatenated to a single vector: $\mathbf{f} = [\mathbf{f}_1...\mathbf{f}_m] \in \mathbb{R}^{F \cdot m}$.

3. $\mathbf{y}_{i-1} \in \mathbb{R}^E$, which is the embedding of the predicted output symbol in the previous decoder step.

Those three inputs are concatenated into a single vector $\mathbf{z}_i = [\mathbf{x}_a, \mathbf{f}, \mathbf{y}_{i-1}] \in \mathbb{R}^{2H+F \cdot m+E}$, which is fed into the decoder, providing the decoder output vector: $\mathbf{LSTM_{dec}}(\mathbf{z}_1...\mathbf{z}_i) \in \mathbb{R}^H$. Finally, to

model the distribution over the possible actions, we project the decoder output to a vector of $|\Sigma_{\hat{y}}|$ elements, followed by a softmax layer:

$$p(s_i = c_j) = softmax_j(\mathbf{R} \cdot \mathbf{LSTM_{dec}}(\mathbf{z}_1...\mathbf{z}_i) + \mathbf{b}) \qquad (4)$$

**Control Mechanism** When the most probable action is $step$, the attention is promoted so $\mathbf{x}_a$ contains the next encoded input representation to be used in the next step of the decoder. This process is demonstrated visually in Figure 1.

## 2.4  TRAINING THE MODEL

For every example: $(x_{1:n}, y_{1:m}, f)$ in the training data, we should produce a sequence of step and write actions $s_{1:q}$ to be predicted by the decoder, which is dependent on the alignment between the input and the output – the network must attend at all the input elements aligned to an output element before writing it. While recent work in sequence transduction advocate jointly training the alignment and the decoding (Bahdanau et al., 2014; Yu et al., 2016), we instead show that in our case it is worthwhile to decouple these stages and learn the hard alignment before hand, using it to guide the training of the encoder-decoder network and enabling the use of correct alignments for the attention mechanism from the beginning of the network training process. For that purpose, we first run a character level alignment process on the training data. We use the character alignment model of Sudoh et al. (2013) which is based on a Chinese Restaurant Process which weights single alignments (character-to-character) in proportion to how many times such an alignment has been seen elsewhere out of all possible alignments. Specifically, we use the implementation provided by the organizers of the SIGMORPHON2016 shared task.[1] Once we have the character level alignment per input-output sequence pair in the training set, we deterministically infer the sequence of actions $s_{1:q}$ that results in the desired output by attending at all the input elements aligned to an output element (using the step action) before writing it. We then train the network to predict this sequence of actions by using a conventional cross entropy loss function per example:

$$\mathcal{L}(x_{1:n}, y_{1:m}, f, \Theta) = -\sum_{s_j \in s_{1:q}} \log softmax_j(\mathbf{R} \cdot \mathbf{LSTM_{dec}}(\mathbf{z}_1...\mathbf{z}_i) + \mathbf{b}) \qquad (5)$$

## 3  EXPERIMENTS

We perform extensive experiments with three previously studied morphological inflection generation datasets to evaluate our hard attention model in various settings. In all experiments we report the results of the best performing neural and non-neural baselines which were previously published on those datasets to our knowledge. The implementation details for the models are available in the supplementary material section of this paper. The source code for the models is available on github.[2]

**CELEX** In order to examine if our model fits the task, we first evaluate it on a very small dataset to see if it avoids the tendency to overfit on few training examples. For this purpose we report exact match accuracy on the German inflection generation dataset compiled by Dreyer et al. (2008) from the CELEX database (Baayen et al., 1993). The dataset includes only 500 training examples for each of the four inflection types: 13SIA→13SKE, 2PIE→13PKE, 2PKE→z, and rP→pA which we refer to as 13SIA, 2PIE, 2PKE and rP, respectively.[3] We compare our model to three competitive baselines that reported results on this dataset: the Morphological Encoder-Decoder (MED) of Kann & Schütze (2016b) which is based on the soft-attention model of Bahdanau et al. (2014), the neural-weighted FST of Rastogi et al. (2016) which uses stacked bi-directional LSTM's to weigh its arcs (WFST), and the model of Dreyer et al. (2008) which uses a weighted FST with latent-variables structured particularly for morphological string transduction tasks (LAT). Following previous reports on this dataset, we use the same data splits as Dreyer et al. (2008), dividing the data for each inflection type into five folds, each consisting of 500 training, 1000 development and 1000 test examples. We train a separate model for each fold and report exact match accuracy, averaged over the five folds.

---

[1] https://github.com/ryancotterell/sigmorphon2016

[2] https://github.com/roeeaharoni/morphological-reinflection

[3] The acronyms stand for: 13SIA=1st/3rd person, singular, indefinite, past;13SKE=1st/3rd person, subjunctive, present; 2PIE=2nd person, plural, indefinite, present;13PKE=1st/3rd person, plural, subjunctive, present; 2PKE=2nd person, plural, subjunctive, present; z=infinitive; rP=imperative, plural; pA=past participle.

**Wiktionary** To neutralize the negative effect of very small training sets on the performance of the different learning approaches, we also evaluate our model on the dataset created by Durrett & DeNero (2013), which contains up to 360k training examples per language. It was built by extracting Finnish, German and Spanish inflection tables from Wiktionary, used in order to evaluate their system based on string alignments and a semi-CRF sequence classifier with linguistically inspired features. We also used the expansion made by Nicolai et al. (2015) to include French and Dutch inflections as well. Their system also performs an align-and-transduce approach, extracting rules from the aligned training set and applying them in inference time with a proprietary character sequence classifier. In addition to those systems we also compare to the results of the recent neural approaches of Faruqui et al. (2016), which did not use an attention mechanism, and Yu et al. (2016), which coupled the alignment and transduction tasks, requiring a beam search decoding procedure.

**SIGMORPHON** As different languages show different morphological phenomena, we also experiment with how our model copes with this variety using the morphological inflection dataset from the SIGMORPHON2016 shared task (Cotterell et al., 2016). Here the training data consists of ten languages, with five morphological system types (detailed in Table 3): Russian (RU), German (DE), Spanish (ES), Georgian (GE), Finnish (FI), Turkish (TU), Arabic (AR), Navajo (NA), Hungarian (HU) and Maltese (MA) with roughly 12,800 training and 1600 development examples per language. We compare our model to two soft attention baselines on this dataset: MED (Kann & Schütze, 2016a), which was the best participating system in the shared task, and our implementation of the global (soft) attention model presented by Luong et al. (2015).

## 4 RESULTS

Table 1: Results over the CELEX dataset

|      | 13SIA | 2PIE | 2PKE | rP   | Avg.  |
|------|-------|------|------|------|-------|
| MED  | 83.9  | 95   | 87.6 | 84   | 87.62 |
| WFST | 86.8  | 94.8 | 87.9 | 81.1 | 87.65 |
| LAT  | **87.5** | 93.4 | 87.4 | 84.9 | 88.3 |
| Hard | 85.8  | **95.1** | **89.5** | **87.2** | **89.44** |

On the low resource setting (CELEX), our model significantly outperforms both the recent neural models of Kann & Schütze (2016b) and Rastogi et al. (2016) and the morphologically aware latent variable model of Dreyer et al. (2008), as detailed in Table 1. It is also, to our knowledge, the first model that surpassed in overall accuracy the latent variable model on this dataset. We explain our advantage over the soft attention model by the ability of the hard attention control mechanism to harness the monotonic alignments found in the data, while also conditioning on the entire output history which wasn't available in the FST models. Figure 2 plots the train-set and dev-set accuracies of the soft and hard attention models as a function of the training epoch. While both models perform similarly on the train-set (with the soft attention model fitting it slightly faster), the hard attention model performs significantly better on the dev-set. This shows the soft attention model's tendency to overfit on the small dataset, as it has significantly more parameters and modeling power and is not enforcing the monotonic assumption of the hard attention model.

Table 2: Results over the Wiktionary datasets

|        | DE-N  | DE-V  | ES-V  | FI-NA | FI-V  | FR-V  | NL-V  | Avg.  |
|--------|-------|-------|-------|-------|-------|-------|-------|-------|
| DDN13  | 88.31 | 94.76 | 99.61 | 92.14 | 97.23 | 98.80 | 90.50 | 94.47 |
| NCK15  | 88.6  | 97.50 | 99.80 | 93.00 | **98.10** | **99.20** | 96.10 | 96.04 |
| FTND16 | 88.12 | **97.72** | **99.81** | 95.44 | 97.81 | 98.82 | 96.71 | 96.34 |
| YBB16  | 87.5  | 92.11 | 99.52 | 95.48 | **98.10** | 98.65 | 95.90 | 95.32 |
| Hard   | **88.87** | 97.35 | 99.79 | **95.75** | 98.07 | 99.04 | **97.03** | **96.55** |

On the large training set experiments (Wiktionary), our model is the best performing model on German verbs, Finnish nouns/adjectives and Dutch verbs, resulting in the highest reported average accuracy across all the inflection types when compared to the four previous neural and non-neural state of the art baselines, as detailed in Table 2. This shows the robustness of our model also with large amounts of training examples, and the advantage the hard attention mechanism provides over the encoder-decoder approach of Faruqui et al. (2016) which does not employ an attention mech-

anism. Our model is also significantly more accurate than the model of Yu et al. (2016), showing the advantage in using independently learned alignments to guide the network's attention from the beginning of the training process.

Table 3: Results over the SIGMORPHON 2016 morphological inflection dataset. The text above each language lists the morphological phenomena it includes: circ.=circumfixing, agg.=agglutinative, v.h.=vowel harmony, c.h.=consonant harmony

|  | suffixing+stem changes | | | circ. | suffixing+agg.+v.h. | | | c.h. | templatic | | |
|  | RU | DE | ES | GE | FI | TU | HU | NA | AR | MA | Avg. |
|---|---|---|---|---|---|---|---|---|---|---|---|
| MED | 91.46 | 95.8 | 98.84 | 98.5 | 95.47 | 98.93 | 96.8 | 91.48 | **99.3** | **88.99** | 95.56 |
| Soft | 92.18 | 96.51 | 98.88 | **98.88** | **96.99** | **99.37** | **97.01** | **95.41** | 99.3 | 88.86 | **96.34** |
| Hard | **92.21** | **96.58** | **98.92** | 98.12 | 95.91 | 97.99 | 96.25 | 93.01 | 98.77 | 88.32 | 95.61 |

As can be seen in Table 3, on the SIGMORPHON 2016 dataset our model performs better than both soft-attention baselines for the suffixing+stem-change languages (Russian, German and Spanish) and is slightly less accurate than our implementation of the soft attention model on the rest of the languages, which is now the best performing model on this dataset to our knowledge.

We explain this by looking at the languages from a linguistic typology point of view, as detailed in Cotterell et al. (2016). Since Russian, German and Spanish employ a suffixing morphology with internal stem changes, they are more suitable for monotonic alignment as the transformations they need to model are the addition of suffixes and changing characters in the stem. The rest of the languages in the dataset employ more context sensitive morphological phenomena like vowel harmony and consonant harmony, which require to model long range dependencies in the input sequence which better suits the soft attention mechanism. While our implementation of the soft attention model and MED are very similar model-wise, we hypothesize that our soft attention results are better due to the fact that we trained the model for 100 epochs and picked the best performing model on the development set, while the MED system was trained for a fixed amount of 20 epochs (although trained on both train and development data).

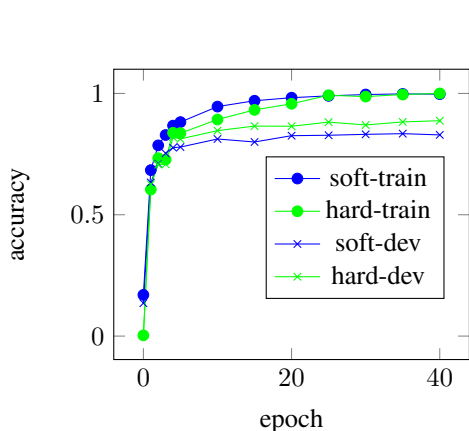

Figure 2: Learning curves for the soft and hard attention models on the first fold of the CELEX dataset

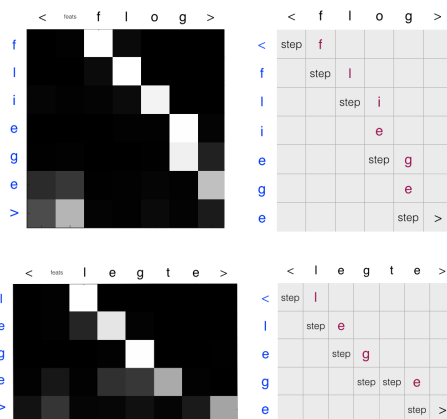

Figure 3: A comparison of the alignments as predicted by the soft attention (left) and the hard attention (right) models on examples from the CELEX dataset.

## 5 ANALYSIS

### 5.1 COMPARISON OF LEARNED ALIGNMENTS

In order to see if the alignments our model predict fit the monotonic alignment structure found in the data, and are they more suitable for the task when compared to the alignments found by the soft attention model, we examined alignment predictions of the two models from the CELEX dataset, depicted in Figure 3. First, we notice the alignments found by the soft attention model are also monotonic, encouraging our modeling approach for the task. We also notice how our model learns to handle morphological phenomena like deletion, as can be seen on the right part of Figure 3, showing

the alignments for the inflection: legte→lege. This inflection requires the model to delete the fourth character of the input sequence. We can see the model learned to delete by reading and writing each character until it reaches the *t* character, which is deleted by performing two consecutive *step* operations. Another notable morphological transformation is the one-to-many alignment, found on the example in the left: flog→fliege, where the model needs to transform a character in the input, *o*, to two characters in the output, *ie*. We can see the model learns to do this by performing two consecutive *write* operations after the *step* operation of the relevant character to be replaced. We also notice that in this case, the soft attention model performs a slightly different alignment by aligning the character *i* to *o* and the character *g* to the sequence *eg*, which is not the expected alignment in this case from a linguistic point of view.

## 5.2 LEARNED REPRESENTATIONS ANALYSIS

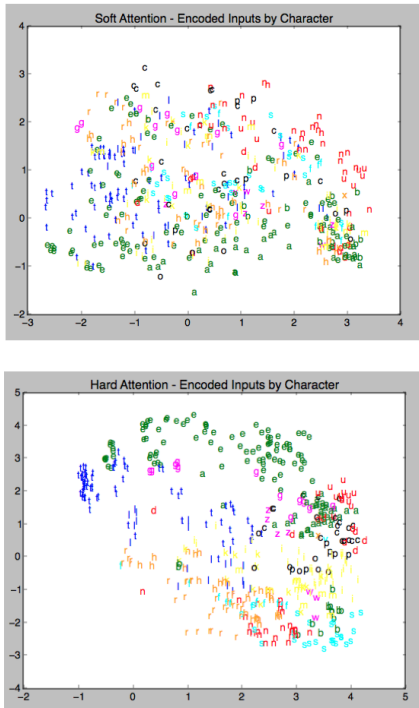 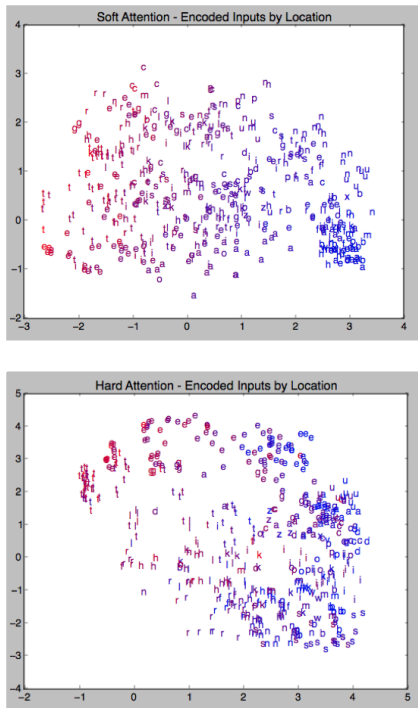

Figure 4: SVD dimension reduction to 2D of 500 character representations in context from the encoder, for both the soft attention (top) and the hard attention (bottom) models. Colors indicate which character is encoded.

Figure 5: SVD dimension reduction to 2D of 500 character representations in context from the encoder, for both the soft attention (top) and hard attention (bottom) models. Colors indicate the location of the character.

When witnessing the success of the hard and soft attention models in sequence to sequence transduction, the following questions arise: how does the models manage to learn monotonic alignments? perhaps the network learns to encode the sequential position as part of its encoding of an input element? In an attempt to answer those questions, we performed the following analysis. We took 500 continuous character representations in context from each model, where every representation is a vector in $\mathbb{R}^{200}$ which is the output of the bi-LSTM encoder of the model. Every vector of this form carries information of a specific character with its context. We perform dimension reduction to reduce those two sets of vectors, each set in: $\mathbb{R}^{500 \times 200}$ into: $\mathbb{R}^{500 \times 2}$ using SVD. We then plot the 2D character-in-context representations and color them in two ways. First, we color the 2D representations by the characters they represent, where we have different color for each character in the alphabet (Figure 4). In the second plot we color the representations by the location of the character they represent in the input sequence: here blue implies the character is closer to the beginning of the sequence and red implies it is closer to the end (Figure 5).

We can see that while both models tend to cluster similar character representations together (Figure 4), the hard attention model tends to have more dense character clusters. This is explained by looking at the location information in Figure 5: while both models encode the positional information to some extent, this information is much more pronounced in the soft-attention mechanism, where the X dimension correlates very strongly with the position information. It seems that the soft-attention mechanism encourages the encoder to encode positional information in its representation. In contrast, our hard-attention model has other means of obtaining the position information in the decoder using the step actions, and indeed does not encode it as strongly in the continuous representations. This behavior allows it to perform well even with fewer examples, as the location information is represented in the network training phase implicitly using the step actions.

## 6  RELATED WORK

Previous works on neural sequence transduction include the RNN Transducer (Graves, 2012) which uses two independent RNN's over monotonically aligned sequences to compute a probability over the possible output symbols in each step, including a null symbol. The model by Yu et al. (2016) improves this approach by replacing the null symbol with a learned transition probability. Both models are trained using a forward-backward approach, marginalizing over all possible alignments. Our model differs from the above by learning the alignments independently, which enables a dependency between the encoder and decoder - the hard attention mechanism. This provided improved results while also greatly simplifying the model training, enabling learning through simple cross-entropy loss. Jaitly et al. (2015) proposed the Neural Transducer model for online speech recognition, which is also trained on external alignments, similarly to our approach. They divide the input into blocks of a constant size and perform soft attention separately on each block when predicting the output symbols. Lu et al. (2016) used a combination of an RNN encoder together with a CRF layer to model the dependencies in the output sequence. A line of work on attention based speech recognition Chorowski et al. (2015); Bahdanau et al. (2016) proposed two relevant improvements to the vanilla attention mechanism: The first adds location awareness by using the previous attention weights when computing the next ones, and the second prevents the model from attending on too many or too few inputs using "sharpening" and "smoothing" techniques on the soft-attention weight distributions.

For the morphological inflection task, previous approaches usually make use of manually constructed Finite State Transducers (Koskenniemi, 1983; Kaplan & Kay, 1994), which require expert knowledge, or machine learning methods (Yarowsky & Wicentowski, 2000; Dreyer & Eisner, 2011; Durrett & DeNero, 2013; Hulden et al., 2014; Ahlberg et al., 2015; Nicolai et al., 2015) with specific assumptions about the set of possible processes that are needed to create the output sequence, requiring feature engineering. More recently, Faruqui et al. (2016) used encoder-decoder neural networks for the task, encoding the input sequence into a vector and decoding it one character at a time into the output sequence. Kann & Schütze (2016a;b) explored the soft attention model proposed for machine translation (Bahdanau et al., 2014) which gave the best results in the SIGMORPHON 2016 shared task (Cotterell et al., 2016). Another notable contribution is the work on weighting finite state transducers with neural context (Rastogi et al., 2016). There, the arcs of an FST are scored by optimizing a global loss function over all the possible paths in the state graph while modeling contextual features with bi-directional LSTM's.

## 7  CONCLUSION

We presented the hard attention model for sequence to sequence transduction of monotonically aligned sequences and evaluated it on the well-studied task of morphological inflection generation. The model employs an explicit alignment model learned independently in training time which is used to teach a neural network to perform both alignment and transduction when decoding with a hard attention mechanism. We showed that our model performs better or on par with more complex soft attention and neural transduction models on various morphological inflection datasets, forming a new state of the art on the CELEX dataset and the Wiktionary dataset and outperforming the best system in the SIGMORPHON2016 inflection generation task. Future work may include experimenting with different external alignment methods, or applying the model to other tasks which require a monotonic align-and-transduce approach like abstractive summarization or transliteration.

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

## SUPPLEMENTARY MATERIAL

### TRAINING DETAILS, IMPLEMENTATION AND HYPER PARAMETERS

To train our models, we used the train portion of the datasets as-is and evaluated the model which performed best on the development portion of the dataset, without conducting any specific pre-processing steps on the data. We train the models for a maximum of 100 epochs over the training set. To avoid long training time, we trained the model for 20 epochs for datasets larger than 50k examples, and for 5 epochs for datasets larger than 200k examples. The models were implemented using the python bindings of the dynet toolkit.[4] We trained the network by optimizing the expected output sequence likelihood using cross-entropy loss as mentioned in equation 5. For optimization we used ADADELTA (Zeiler, 2012) without regularization. We updated the weights after every example. We used the dynet toolkit implementation of an LSTM network with two layers, each having 100 entries in both the encoder and decoder. The character embeddings were also vectors with 100 entries for the CELEX experiments, and with 300 entries for the SIGMORPHON and Wiktionary experiments. The morpho-syntactic attribute embeddings were vectors of 20 entries in all experiments. We did not use beam search while decoding for both the hard and soft attention models as it is significantly slower and did not show clear improvement in previous experiments we conducted. In all experiments, for both the hard and soft attention models, we report results using an ensemble of 5 models with different random initializations by using majority voting on the final sequences the models predicted, as reported in Kann & Schütze (2016b). This was done to perform fair comparison to the models of Kann & Schütze (2016b;a); Faruqui et al. (2016) which also performed a similar ensembling technique.

---

[4]https://github.com/clab/dynet

