# Peer review of "Sequence to Sequence Transduction with Hard Monotonic Attention"

_ICLR 2017 — rejected_

[Public Comment · Jason Eisner · 29 Nov 2016]
**typos in section 2.3 "Encoder"**

In the "Encoder" section, the authors appear to use plain math italic x_i for input elements (characters) and boldface x_i for biLSTM encodings.

I believe the subscript in e_{x_i} should be the plain version since here x_i represents an input element.
Also, the definition of boldface x_i should take e_{x_i} and not boldface x_i as input; otherwise the definition is circular.

(The numeric subscripts i should not be boldfaced either since they are also not vectors; but that typo is less confusing.)

Correct?

[Official Review · AnonReviewer1 · rating 4 · confidence 5 · 16 Dec 2016]
**Not novel enough**

The paper proposes an approach to sequence transduction for the case when a monotonic alignment between the input and the output is plausible. It is assumed that the alignment can be provided as a part of training data, with Chinese Restaurant process being used in the actual experiments. 

The idea makes sense, although its applicability is limited to the domains where a monotonic alignment is available. But as discussed during the pre-review period, there has been a lot of strongly overlapping related work, such as probabilistic models with hard-alignment (Sequence Transduction With Recurrent Neural Network, Graves et al, 2012) and also attempts to use external alignments in end-to-end models (A Neural Transducer, Jaitly et al, 2015). That said, I do not think the approach is sufficiently novel. 

I also have a concern regarding the evaluation. I do not think it is fair to compare the proposed model that depends on external alignment with the vanilla soft-attention model that learns alignments from scratch. In a control experiment soft-attention could be trained to match the external alignment. Such a pretraining could reduce overfitting on the small dataset, the one on which the proposed approach brings the most improvement. On a larger dataset, especially SIGMORPHON, the improvements are not very big and are only obtained for a certain class of languages.

To sum up, two main issues are (a) lack of novelty (b) the comparison of a model trained with external alignment and one without it.

[Official Review · AnonReviewer3 · rating 5 · confidence 4 · 17 Dec 2016]
**Nice idea, but limited applicability (need an auxiliary solver for alignments)**

The paper describes a recurrent transducer that uses hard monotonic alignments: at each step a discrete decision is taken either to emit the next symbol or to consume the next input token. 

The model is moderately novel - similar architecture was proposed for speech recognition (

[Official Review · AnonReviewer2 · rating 5 · confidence 3 · 17 Dec 2016]

This paper proposes a sequence transduction model that first uses a traditional statistical alignment methods to provide alignments for an encoder-decoder type model. The paper provides experiments on a number of morphological inflection generation datasets. They shows an improvement over other models, although they have much smaller improvements over a soft attention model on some of their tasks. 

I found this paper to be well-written and to have very thorough experiments/analysis, but I have concerns that this work isn't particularly different from previous approaches and thus has a more focused contribution that is limited to its application on this type of shorter input (the authors "suggest" that their approach is sufficient for shorter sequences, but don't compare against the approach of Chorowski et al. 2015 or Jailty el at 2016).

In summary, I found this paper to be well-executed/well-written, but it's novelty and scope too small. That said, I feel this work would make a very good short paper.

[Final Decision · Program Chairs · 06 Feb 2017]
**ICLR committee final decision**

While this area chair disagrees with some reviewers about (1) the narrowness of the approach's applicability and hence lack of relevance to ICLR, and also (2) the fairness of the methodology, it is nonetheless clear that a stronger case needs to be made for novelty and applicability.